# Toxicological Aspects Associated with Consumption from Electronic Nicotine Delivery System (ENDS): Focus on Heavy Metals Exposure and Cancer Risk

**DOI:** 10.3390/ijms25052737

**Published:** 2024-02-27

**Authors:** Silvia Granata, Fabio Vivarelli, Camilla Morosini, Donatella Canistro, Moreno Paolini, Lucy C. Fairclough

**Affiliations:** 1Department of Pharmacy and Biotechnology, Alma Mater Studiorum-University of Bologna, Via Irnerio 48, 40126 Bologna, Italy; silvia.granata3@unibo.it (S.G.); fabio.vivarelli3@unibo.it (F.V.); camilla.morosini2@unibo.it (C.M.); moreno.paolini@unibo.it (M.P.); 2School of Life Sciences, University of Nottingham, East Dr, Nottingham NG7 2TQ, UK; lucy.fairclough@nottingham.ac.uk

**Keywords:** ENDS, cancer, oxidative stress

## Abstract

Tobacco smoking remains one of the leading causes of premature death worldwide. Electronic Nicotine Delivery Systems (ENDSs) are proposed as a tool for smoking cessation. In the last few years, a growing number of different types of ENDSs were launched onto the market. Despite the manufacturing differences, ENDSs can be classified as “liquid e-cigarettes” (e-cigs) equipped with an atomizer that vaporizes a liquid composed of vegetable glycerin (VG), polypropylene glycol (PG), and nicotine, with the possible addition of flavorings; otherwise, the “heated tobacco products” (HTPs) heat tobacco sticks through contact with an electronic heating metal element. The presence of some metals in the heating systems, as well as in solder joints, involves the possibility that heavy metal ions can move from these components to the liquid, or they can be adsorbed into the tobacco stick from the heating blade in the case of HTPs. Recent evidence has indicated the presence of heavy metals in the refill liquids and in the mainstream such as arsenic (As), cadmium (Cd), chromium (Cr), nickel (Ni), copper (Cu), and lead (Pb). The present review discusses the toxicological aspects associated with the exposition of heavy metals by consumption from ENDSs, focusing on metal carcinogenesis risk.

## 1. Introduction

Although there has been a significant decrease in the overall tobacco consumption in several countries, smoking still remains one of the main health issues worldwide [1], making smoking cessation a critical world health priority [2]. Multi-level campaigns to discourage this habit aim to prevent non-smokers from starting smoking and encourage active smokers to stop smoking [3], but most smoking-cessation attempts end in failure, with less than 10% of smokers successful for at least 12 months [4]. Electronic Nicotine Delivery Systems (ENDSs) give the possibility to inhale nicotine, maintaining the typical gestures associated with tobacco smoking, without the exposure to whole toxic combustion by-products, and it is little wonder that it has received great attention as a putative aid to stop smoking [5]. Although e-cigarette (e-cig) companies continually release innovative products onto the market, two main classes of these devices are identified: “liquid e-cigs” that include an atomizer that vaporizes a liquid composed of vegetable glycerin (VG), polypropylene glycol (PG), and nicotine with the possible addition of flavorings, and the “Heat-not-Burn systems” (HnB) or “Heated Tobacco Products” (HTPs) that heat tobacco sticks through contact with an electronic heating metal blade [5]. Recent evidence showed that both liquid e-cigs and Heat-not-Burn systems release toxic compounds such as aldehydes, carbonyls, and free radicals mainly derived from the thermal degradation of VG and PG [6].

The ENDS systems contain elements that include the same harmful constituents of conventional cigarettes (CCs), such as polycyclic aromatic hydrocarbons (PAHs), aldehydes, and carbonyls derived from incomplete pyrolysis and the thermogenic degradation of tobacco [7]. Heavy metals have also been detected in the ENDS mainstream [8,9,10], raising significant concerns for their potential oncogenic role. The present review discusses the toxicological aspects associated with the exposition to heavy metals by consumption from ENDSs, focusing on cancer risk.

## 2. The ENDS Devices

Worldwide, the tobacco epidemic has caused the death of more than 8.7 million people per year, among which more than seven million are caused by direct tobacco use, whilst around 1.2 million are caused by second-hand smoke in non-smokers. It is commonly known that tobacco smoke is one of the risk factors for several diseases that can be easily removed [2], such as respiratory and cardiovascular disease, as well as cancer [11]. Its impact on chronic diseases is well proven and undeniable [1]; hence, in the 1950s, tobacco industries made efforts to reduce the toxicants, especially the overall tar and nicotine yields of CCs, through the introduction of filters, to try to decrease their negative effects [12]. 

Unfortunately, those changes were not effective, as tobacco smokers found ways to compensate for them; for instance, smokers covered the ventilation holes introduced in CCs so that smoke exposure was no longer reduced [13,14]. Furthermore, it became clear there was no direct relationship between tar and nicotine levels and risks for CC-related diseases [15]. In addition, lower nicotine or tar contents cannot be translated into a risk reduction, since nicotine addiction leads smokers to consume a great number of “light cigarettes” [16]. 

As modifications to CCs were unsuccessful, the World Health Organization (WHO) proposed guidelines to discourage the use of tobacco, including the use of pictorial health warnings on the packaging, smoke-free laws to protect the health of non-smokers, bans on tobacco advertising to reduce consumption, and the introduction of tobacco taxes [2]. The success of these measures has been shown in several studies, which have revealed that significant reductions in smoking have been caused by higher taxes on CCs and that restrictions on sponsorship and smoking indoors and in the workplace also showed advantages [17,18]. 

With WHO oversight, in 2005, the world’s first international treaty for the protection of public health was established and named “Framework Convention on Tobacco Control”. It aims to reaffirm the right to the highest standards of health for all people, and to do so, it challenges some of the causes of the tobacco epidemic previously described. The Framework Convention establishes the objectives and legally binding principles that the signatories (countries or organizations for economic integration, such as the European Union) are required to respect [19,20].

### 2.1. E-Cigs

Liquid e-cigs are battery-operated devices that release nicotine by heating a solution containing vegetable glycerin (VG) and propylene glycol (PG). In the following, we refer to emissions from different devices as “mainstream”. The typical components of an e-cig include a mouthpiece, a cartridge/tank holding the e-liquid, an atomizer or heating coil that aerosolizes the e-liquid, a battery, and a sensor or user-actuated button to activate the atomizer [21] (Figure 1). E-cigs conventionally vaporize the e-liquid through an atomizer powered by a rechargeable lithium battery. The most appealing feature is the possibility to customize these devices with different PG/VG ratios combined with the possibility of choosing from a wide range of flavored liquids with different nicotine concentrations [22]. The standard power supply voltage is 3.7 V, although on the market, it is possible to find models called “BB” (big battery) which allow a greater heating of the liquid to make the aerosol denser and more concentrated. It is possible to distinguish two main classes of e-cigs: “closed” e-cigs, which cannot be refilled, nor can any fundamental part of the device (battery or atomizer) be replaced by the customer; or “open” systems, which allow the users to customize the device, namely by replacing several components such as the atomizer (single or multiple coil) with a range of electrical resistances in terms of Ohms, voltage applied, and composition of the e-liquid with more than 15,586 e-liquid flavors reported [23,24,25,26,27]. 

The voltage of the battery can be regulated by the users through a large selection of atomizers with electrical resistance values ranging from sub-Ohm (<0.5 Ω) up to >2.8 Ω coils [28]. The low voltage was perceived as safer; however, in vitro models have demonstrated that the use of sub-Ohm atomizers leads to a lower cell viability and higher levels of reactive oxygen species [29]. In addition, data from in vivo models indicate that the use of sub-Ohmic coil resistance is associated with a greater impairment of lung tissue [22,30]. Again, nicotine-free liquids, initially advertised as almost risk-free, can induce some changes in gonadal function that are typically observed in smoking models. In fact, in an exposed group of alveolar collapse, a reduction in ciliated cells in the trachea and the presence of necrotic and apoptosis bodies was observed; however, these alterations were more evident in the 1.5 Ω group than in the 0.25 Ω group [22,29,30]. 

The composition of the e-liquid can vary in terms of the percentage of VG, PG, and nicotine, with the latter usually ranging from 0 to 36 mg/mL, but it can reach a concentration of 90 mg/mL in some cases. The level of nicotine is of great relevance since it can affect plasma nicotine levels. The lower temperature reached by e-cigs promotes the dehydrogenation of nicotine into nicotyrine, which causes the inhibition of cytochrome P450 isoforms and increases the levels of nicotine in the serum [31]. In addition, the ratio of PG and VG can impact the concentration of nicotine and nicotyrine in e-cigs: higher nicotine concentrations were found in PG-based e-liquids than VG-based e-liquids [31,32], as well as more toxic chemicals including formaldehyde, acrolein and acetaldehyde, polycyclic aromatic hydrocarbons (PAHs), as well as volatile organic compounds (VOCs), which are classified as carcinogenic to humans [33,34,35]. Furthermore, it has been reported that the chemical transformation of flavors may generate free radical species [36,37,38]. 

### 2.2. Heated Tobacco Products (HTPs)

HTPs or Heat not-Burn (HnB) tobacco products are hybrids between CCs and e-cigs. CCs volatilize nicotine through tobacco combustion, at temperatures around 700 °C. HTPs use electronics to heat tobacco, but at significantly lower temperatures than CCs (about 300 °C) [39]. HTPs consist of a tobacco stick (designated HEETS or HeatStick), a battery-operated holder which heats tobacco, and a charger where the holder is kept and recharged after each use (Figure 1). HeatSticks are composed of ground tobacco that is reconstituted into sheets with water, cellulose fibers, guar gum, and glycerin, which are then fashioned into small plugs. The heater is composed of platinum, gold, silver, and a ceramic coating [40,41]; moreover, the emission of some substances depends on the device cleaning strategy [42]. In fact, as discussed below, the use of a metal blade as a tobacco heating element in the case of HTPs normally releases significant amounts of debris, fluids, and fragments of cast left in the holder and, for this reason, it is necessary to clean the devices after 20 tobacco sticks. The most common devices use the blade to heat the tobacco sticks; however, new heating systems have been developed, such as an induction heating chamber (Glo^TM^), where tobacco sticks are placed inside the device, and the use of an external resistive heater made of stainless-steel tracks is utilized [43].

Data from manufacturers report a 90% to 95% reduction in the toxic chemicals compared to CC smoke, and some independent studies confirmed the results. However, harmful chemicals are not absent and, in fact, have been suggested to emit more PAHs, carbonyls, and submicron particles than e-cigs [44,45,46,47,48,49].

In addition, the number, duration, and volume of puffs are also suggested to affect the exposure to toxic chemicals, such as carbonyls. A more intense puffing regiment was found to enhance the inhalation of formaldehyde, acetaldehyde, crotonaldehyde, or propionaldehyde [50]. 

## 3. Heavy Metals in Tobacco, CC Smoke, E-Cig, and HTP Mainstream

Heavy metals are hazardous compounds for human health and might lead to pathologic conditions, as well as cancer [51]. It is known that some metals are more associated with health issues than others, such as arsenic (As), cadmium (Cd), chromium (Cr), nickel (Ni), copper (Cu), and lead (Pb) [52,53]. Although the precise mechanisms of oncogenesis remain to be fully elucidated, some of the postulated ones include oxidative stress by means of free-radical generation (at the level of the electron transport chain in the mitochondrion), direct genotoxicity by metals/metal ions, and alterations in stem cell function/gene expression [10]. A putative metal contamination during the ENDS manufacturing processing or an endogenous mobilization from the heating elements were hypothesized from the early years of their market launch. Since the heating element is usually made of a metal, and other metals are also frequently present, including solder joints, and considering the temperatures along with the concurrent generation of volatile organic compounds, heavy metal ions can move from these components to the liquid, or they can be adsorbed onto the tobacco stick. Again, considering the significant number of organic compounds present in ENDSs, such as aldehydes and ketones, the formation of complexes with metals ions is highly likely [40,41]. As recently reviewed [10], CCs report an average metal content (expressed as μg/g of tobacco dry weight) around 0.14 for As, 0.79 for Cd, 1.39 for Cr, 2.10 for Ni, and 0.55 for Pb. The rate of metal transfer from tobacco to CC smoke (expressed as percentage of metals migrating from tobacco matrix to smoke) ranges from 33 to 44% for As, 69 to 85% for Cd, 20% for Cr, 56.3 to 65.5% for Ni, and 54.3 to 67.1% for Pb [52]. A recent study reports the following concentrations for e-cigs expressed in ng/m^3^: 6.86 for Cr, 0.01 for Cd, 0.30 for Ni, and 0.19 for Pb [54]. HTPs present concentrations <0.36 μg/stick for As, <0.28 μg/stick for Cd, <15.9 μg/stick for Ni, and 2.23 μg/stick for Pb [51]. These data are summarized in Table 1.

### 3.1. E-Cigs

Several studies have reported the presence of all the heavy metals mentioned above in the e-cig mainstream, as well as in the refill liquids used with such devices [8,9]. Hess and colleagues analyzed, with Inductively Coupled Plasma Mass Spectrometry (ICP-MS), ten refill liquids for five different brands of e-cigs, and they reported the presence of Cd, Cr, Pb, Ni, and manganese (Mn), with large concentration ranges among the liquids examined (3.2–1960 μg/L for Pb, 53.9–2110 μg/L for Cr, 115–22,600 μg/L for Ni, 28.7–6919 μg/L for Mn) [8]. Similarly, Kamilari detected heavy metals in the twenty refill liquids studied through Total Reflection X-Ray Fluorescence Spectroscopy (TXRF), even though they highlighted that the concentrations were below the limit defined by the regulatory authorities and lower than what was reported by Hess [9]. Another study, conducted by Olmedo and colleagues using ICP-MS, shows evidence of heavy metals in e-liquids of e-cigs. Specifically, this study found eleven metals with a wide range of concentrations in the tank samples, as well as in the mainstream generated by the devices [55]. Their concentrations, as well as those detected by Kamilari, were considered significantly lower than the ones found in CCs. In particular, Pb, Ni, and Cr appear on the FDA’s “harmful and potentially harmful chemicals” list [56] and, notably, Pb and Cr concentrations in e-cigs were within the range of CCs, while Ni likely originated from the nichrome wire, and some particles in the wire were about 2–100 times higher in the e-cig mainstream than in Marlboro brand CCs [57]. However, since e-cigs vaporize a liquid, one of the main variables in the presence or not of heavy metals is, indeed, the liquid composition [58]. 

### 3.2. HTPs

Concerning HTPs, only a few research studies have been conducted by investigating the content of heavy metals in the sticks. In 2020, Koutela and colleagues showed that Cr, Cu, Ni, selenium (Se), Cd, mercury (Hg), and Pb are present in both used and unused HTP sticks in a significantly lower concentration than CCs [59]. However, this study did not discuss the rate of metals transferred in the mainstream. The study conducted by Amorόs-Pérez and colleagues found in the HTP mainstream that mainly organic compounds (C and O) were identified, and these are usually also recorded in CC smoke. Moreover, in HTPs, they detected using ICP-MS the presence of As (0.05 ppb) in a much lower concentration when the ratio of its concentration in CCs and in HTPs was calculated (1.33 ratio Ccig/CHTP) [60]. So far, our research group published data on the chemical analysis of the HTP mainstream, investigating the presence of heavy metals as well, and Cu was detected at a concentration of 350 ± 53 µg/stick [61]. 

Interestingly, in HTPs, the presence of α- and β- particles, namely polonium-210 (210Po) and lead-210 (210Pb), has recently been identified [62,63]. These radioactive particles are naturally present on tobacco leaves [64], and their presence in tobacco smoke has been demonstrated by Little et al. [65,66,67], as well as their preferential localization in the bronchial epithelium of smokers. Some studies indicate that the level of radioactive Pb released by HTP was higher than in CC smoke, probably due to the humectant glycerol used in HTP that could be more efficient in carrying 210Pb-bearing particles at lower temperatures than CCs [63]. Worryingly, even though, in 1974, Little and O’Toole [67] found proof that these particles cause lung cancer, the tobacco industry has done nothing to remove them from tobacco [64] and it is concerning how their presence persists in these new device sticks. Furthermore, as mentioned above, the HTP mainstream contains elements from pyrolysis and thermogenic degradation that are the same harmful constituents of CC smoke, such as aldehydes and PAHs, as recently reported [7] and summarized in Table 2.

## 4. Metals and Molecular Carcinogenesis

The International Agency for Research on Cancer (IARC) has classified tobacco smoking as carcinogenic to humans (classified as group 1) [68]. The smoke produced by tobacco combustion in CCs is a “concentrate of liquid particles suspended in an atmosphere consisting mainly of nitrogen, oxygen, carbon monoxide and carbon dioxide” [69]. Innumerable studies have examined the toxicity of individual chemicals found in smoke, identifying more than 70 as carcinogens, such as aromatic amines, PAHs, and, in particular, some heavy metals [40,52]. Among these harmful compounds, the metals seem to be crucial for the toxicity of tobacco smoke, with metal exposure potentially increasing the risk of cancer in the users at different sites.

Currently, the potential danger of e-cigs is challenging to define, due to the wide variability and constant modification of existing devices, resulting in differences in engineering and ingredients, in addition to the users’ possibility of personalizing the vaping experience, as previously described. Given that the range of metals is variable across the e-cig brands [8,70], Cr and Ni are the most abundant in the e-cig mainstream; in particular, these metals occur in higher concentration than in CC smoke due to their leaching from the core assembly [71]. This is especially relevant since Ni and Cr are classified in group 1 carcinogens for humans by IARC [72].

Due to the recent introduction of ENDS on the market, knowledge of their long-term effects is limited; however, in vitro and in vivo studies have reported an increased risk of cancer development [5]. The first in vitro studies demonstrated that human bronchial epithelial cells, exposed to the e-cig mainstream, showed gene expression pathway changes similar to those elicited by conventional smoke, supported by the observation of malignant transformation features in airway epithelial cells exposed to e-cig emission [73]. Moreover, e-cigs could promote the epithelial–mesenchymal transition (EMT) and the release of oncogenic cytokines or microRNA from pulmonary and breast cancer cells, thus supporting the association between ENDS use and cancer [74,75,76].

In vivo studies showed that e-cig exposure promotes carcinogenesis by the upregulation of the carcinogen-metabolizing enzymes and inducing oxygen free radical production, leading to oxidative damage to macromolecules including lipids, proteins, and DNA. Furthermore, the results of mutagenesis tests supported the hypothesis that the use of e-cigs can promote genotoxic effects [5]. In this regard, a critical role is played by the presence of heavy metals in the ENDS mainstream (Figure 2).

It has been shown that the presence of metals in e-cigs could contribute to a five-fold increase in lung cancer for smokers versus nonsmokers [77,78]. It has also been suggested that e-cig use could contribute to the development of basaloid squamous cell carcinoma (BSCC) in humans, after using e-cigs every day for 13 years [79], and could cause severe lung and liver inflammation, stimulating metastatic disease by promoting epithelial–mesenchymal transition translocation and interfering with DNA repair mechanisms [75]. Moreover, different aromatic amines and aldehydes, among other bladder carcinogens, have been found higher in the urine of e-cig users when compared to non-smokers [80]. In fact, a case report showed that many carcinogens in ENDS can be metabolized into different compounds that could induce lung or bladder malignancies or heart disease [81,82].

In addition, exposure to these devices also enhances the development of breast cancer and metastasis, through the regulation of myeloid cells; in fact, a state of chronic inflammation results in a tumor-promoting microenvironment, promoting a progression of tumor and also lung metastasis of human breast cancer cells [83,84]. Interestingly, Cd and Ni could act as metalloestrogens and play a critical role in breast cancer development. Epidemiological studies have found higher concentrations of Cd in the tissue, blood, and urine of patients with malignant breast cancer than those with benign tumors [85]. In particular, Erα-positive breast cancers had higher Cd concentrations than Erα-negative cancers; in fact, Cd is able to mimic the estrogens and active Erα [86,87].

The precise mechanism of ENDS-induced cancer is not completely understood. Below, the main hypothesized mechanisms and the contribution of metals are discussed (Figure 3).

### 4.1. Metal Carcinogenesis Mechanisms: Oxidative Stress and DNA Damage

It is known that the increased production of reactive oxygen species (ROS) and the consequent higher levels of oxidative stress contribute to the development of tumorigenic processes [88,89]. Although the exact molecular mechanism of metal-induced carcinogenesis remains unclear, a vast body of evidence indicates that the metal-induced generation of oxidative stress may play a central role in this process [90]. Metal ions for As, Cr, Ni, Cu, and iron (Fe) are capable of interfering with redox reactions in biological systems and they are considered an important source of ROS where the highly reactive hydroxyl radical (•OH) is generated via Fenton and/or Haber–Weiss reactions by the breakdown of H_2_O_2_ [77]. The superoxide anion and hydrogen peroxide are two types of reactive oxygen species that are created when human keratinocytes are exposed to As. This pathway is crucial for As-induced carcinogenesis: the reaction between ROS and DNA leads to the formation of 8-hydroxy-29-deoxyguanosine (8-OHdG), which can cause G > T conversions, which can lead to G > C → T > A transversions that are associated with As-related carcinogenesis [91]. Furthermore, Pb can cross-link proteins and DNA, attach to the phosphate groups of DNA, and modify its shape in vitro, but only at relatively high Pb/DNA ratios. In vitro studies showed how Pb exposure can cause chromosomal abnormalities, micronuclei, sister chromatid exchanges, and, in some studies, point mutations [92]. Cd has been found to generate oxidative stress by inhibiting the antioxidant enzyme machinery such as catalase, superoxide dismutase, and glutathione peroxidase [93], whereas Cr, As, and Ni result in the generation of a whole spectrum of ROS, including the superoxide radical (O2•^−^), hydrogen peroxide (H_2_O_2_), and the hydroxyl radical (•OH), which appear to play a role in malignant transformation [77]. Exposure to metals can affect the normal balance between ROS and antioxidant defenses, resulting in an excessive ROS production and subsequent damage to lipids, proteins, and DNA. On the other hand, reduced oxidative stress can also act as signaling molecules in the maintenance of antioxidant and antiapoptotic conditions, a process called apoptosis resistance, resulting in a lower oxidative stress environment that enables tumor cell survival and proliferation [94].

The exposure of both e-cigs and HTPs is associated with increased levels of mitochondrial ROS due to the disruption of the electron transport chain in vitro [95]. This is confirmed by in vivo studies that show boosted reactive radicals, especially in the lung, liver, and testis [30,44,61]. In addition, several studies have reported pro-inflammatory effects, similar to those induced by CCs. The over-production of ROS leads to an activation of NF-κB, a transcriptional factor involved in the regulation of many pro-inflammatory genes, including cytokines and chemokines [96,97]. These results are associated with a reduction in glutathione (GSH), one of the most important scavengers of reactive species, and increased levels of antioxidant response element (e.g., NRF2, interleukin (IL)-6, and TNf-α) production [96,98].

Intriguingly, NRF2 induction, which would neutralize further oxidative stress and restore the cellular redox balance, appears to be beneficial during premalignant states in which cells have not yet reached a level of DNA damage that makes them malignant. On the contrary, NRF2 becomes undesirable in the late stage due to the ability of constitutively activated NRF2 to ultimately play an oncogenic role and cause malignant cancer cells resistant to apoptotic cell death and/or treatments. As-, Cd-, Cr-, and Ni-transformed cells show NRF2 constitutively activated with an upregulation of NRF2 antioxidant and antiapoptotic target genes [95]. This leads to a decrease in ROS level, maintaining a favorable environment for the progression of transformed cells and tumor formation [99]. Furthermore, the proinflammatory effects associated with e-cig exposure can also be attributable to metal presence since Cr- and As-transformed cells exhibit higher levels of COX-2, TNF-α, and NF-κB due to the constitutive activation of NRF2 that regulates the expression of these inflammatory mediators [100].

Additional studies have evaluated the toxicological outcomes derived from e-cig use, both in vitro and in vivo, highlighting their mutagenic potential. In vitro systems revealed cell cytotoxicity and genotoxicity triggered by e-cigs [101,102]. Inflammation is known to aggravate DNA damage in a synergistic way, which can result in mutations and altered proteins and, ultimately, increase the risk of cancer development [103]. ENDSs release compounds able to induce DNA damage, as single- or double-stranded breaks, DNA fragmentation, and mutations [81,104]. Data from in vivo models have shown a decrease in the expression of XPC and OGG1/2 in lung tissue, key enzymes involved in DNA nucleotide and base excision repair [81]. Additionally, exposed cell lines expressed increased H2A histone family member X (-H2AX), a recognizable indicator of double-stranded DNA breakage [105].

In this regard, the metals found in the e-cig mainstream, possibly by contamination from the metal components of devices, such as the battery and the atomizer, may contribute to DNA damage and mutagenesis [106,107]. However, most metals, like Cd and As, are not direct mutagens. In fact, the induction of oxidative stress by metal ions (such as Cr, Ni, and As) could explain the mutagenic and carcinogenic effects of these metals [77]. Data from cell models have shown an increased production of ROS after exposure to different metals, including the superoxide radical and hydrogen peroxide, in human lung bronchial epithelial cells (BEAS-2B) [108,109]. Moreover, metal ions can inhibit the repair of DNA damage, through the interference with base and nucleotide excision repair and with the repair of stable DNA products. Interestingly, data from in vitro models of exposure to different brands of e-cigs with or without nicotine, as well as heavy metals (like Ni and Pb), showed decreased cell viability and apoptosis compared to the control group [105].

### 4.2. Epigenetic Mechanisms and Gene Expression Regulation

Epigenetic modifications of the genome commonly include DNA and RNA methylations, histone modifications, and non-coding RNAs. These modifications regulate gene expression by altering the local structural dynamics of chromatin, primarily regulating its accessibility and compactness [110]. DNA methylation and demethylation are dynamic processes, which harbor a fundamental function in regulating gene expression. In general, DNA methylation leads to gene silencing by inhibiting the transcriptional factors from binding to DNA, remodeling chromatin to make it less accessible [111]. In lung cancer tissues of mice exposed to As, a high prevalence of promoter hypermethylation of the tumor suppressors p16INK4A and RASSF1A was observed [112]. Since aberrant DNA promoter hypermethylation is significantly associated with the silencing of cancer-promoting genes, this could represent a further carcinogenesis mechanism linked to As exposure. Consistently, the malignant transformation of human non-tumorigenic cell lines by As compounds has also been linked to DNA promoter methylation, and decreased expression of the DBC1, FAM83A, ZSCAN12, and C1QTNF6 genes [113]. Again, As can also cause interference with microtubule assembly, increasing the number of micronuclei, which has been associated with new cancer occurrences [114]. Aberrant H3K9 and H3K27 methylation patterns are associated with various forms of cancer [115]. Cr exposure caused globally increased levels of H3K9me2/3 in human lung cancer cells. Cr also induces an increment in H3K9me2 in the promoter tumor suppressor MLH1, contributing to Cr-associated carcinogenesis [10]. Also, Ni significantly increases global levels of H3K9me2 and H3K4me3 [116]. H3K27me3 is involved in Ni-induced in vitro EMT through the irreversible activation of ZEB1 as the master EMT driver [116].

Emerging evidence indicates how the activation of some stress response elements, such as metallothionein (MT) and heat shock proteins (HSPs), plays a key role in Cd carcinogenesis [117]. Furthermore, exposure to Cd can increase the expression of several translation factors including the eukaryotic translation initiation factor 4E (eIF4E) and eukaryotic translation elongation factor 1A2 (eEF1A2). These elements might contribute to the onset of cancer [117].

### 4.3. microRNAs

MicroRNAs (miRNAs) are large groups of small non-coding RNAs that negatively regulate the coding protein expression. Several studies have shown that non-coding RNAs are dysregulated in cells exposed to heavy metals and play important roles in metal-induced carcinogenesis [111]. As recently reviewed, Hg Cd, and As exposure significantly affects microRNA expression levels in vitro and in vivo in different tissues, and some of these can play a key role in tumorigenesis. In particular, miR-190 expression increases in response to As exposure, resulting in the downregulation of PH domain and Leucine rich repeat Protein Phosphatases (PHLPP) and Tumor Protein P53-Inducible Nuclear Protein 1 (TP53INP1), two essential tumor suppressors [118]. A reduced PHLPP expression results in an extended Akt target gene activation including vascular endothelial growth factor (VEGF) that correlates with malignant transformation [118]. As, Cr, and Ni can upregulate miR-21, promoting angiogenesis and increasing the cell invasive potential [119], and can directly downregulate programmed cell death 4 (PDCD4), as a tumor suppressor [120]. Ni exposure induced miR-21 expression via activating the epidermal growth factor receptor EGFR/NF-κB signaling pathway; upregulated miR-21, in turn, suppressed the expression of two miR-21 target genes, SPRY2 and reversion-inducing cysteine-rich protein with kazal motifs (RECK), and promoted lung cancer cell invasiveness [10]. Cd exposure also impairs miRNA expression, including the upregulation of miR-124-3p and miR-370-3p that promote Cd-induced apoptosis by directly targeting Bcl-2 as an apoptosis suppressor [10].

### 4.4. EGFR and MAP Kinase

The EGFR is implicated in the carcinogenesis of different types of cancer, such as lung cancer. The overexpression ranges from 43% to 89% in non-small cell lung cancer [121]. It has been reported that patient-derived glioblastoma stem-like cells exposed to e-liquids showed an increased level of phospho-EGFR, leading to autophagy deficiency and to the activation of ERK [122,123]. In addition, in vivo studies showed that the injection of patient-derived brain tumor cells into mice brains accelerated tumor growth and the poor prognosis in the e-liquid-treated group compared to the control group [94,95]. In addition, the exposure to Cr activates EGFR in the lung tissue of animals, as well as in non-smoking people exposed to Cr [77,124].

Recently, it was shown that MAP kinase pathways (ERK1/2, JNKs, p38) are involved in tobacco-smoke-induced EMT, a critical regulator of cancer initiation and development [125]. EMT induction results in the downregulation of epithelial cadherin, loss of cell–cell adhesion, as well as increased mobility of cells, all identified as early events in carcinogenesis. The exposure to ENDS can activate the MAPK pathways, resulting in the increased phosphorylation of ERK, both in cell lines and animals exposed to these devices [61,126]. In addition, the exposure to Ni, Cr, and Cd induced the activation of this latter signaling pathway [127,128], suggesting that the presence of heavy metals in the ENDS mainstream could participate and contribute to the alteration in cancer molecular markers.

### 4.5. c-Myc and Cell Cycle

Many heavy metals, such as Ni, Cd, and Cr, are able to alter cell apoptosis, in addition to the activation of some oncogenes. The exposure to Cd alters the expression of immediate early genes, in particular c-Myc, leading to cellular proliferation. The resistance to apoptosis is determined by the mutations of genes, such as BCL-2, p53, and BAX, and it can contribute to the Cd-malignant transformation of epithelial cells [129,130]. Many studies have reported higher levels of c-Myc in tumorigenic cells, while low levels have been found in normal proliferating cells. The exposure of normal BEAS-2B cells to nickel chloride (100 μM) resulted in an induction of c-Myc and an increased number of anchorage-independent colonies, suggesting that Ni ions may promote BEAS-2B cell transformation [131].

## 5. Exposure to Metals Delivered from ENDS and Lung Cancer Risk

The lung is the main target organ of volatile toxicants and there is growing concern that e-cig consumption may increase the risk of lung cancer, although the relationship is still not well established, and the long-term effects will take years to develop [132]. The putative oncogenicity associated with e-cig consumption has been attributed to several distinct molecular pathways, including oxidative stress, the epithelial–mesenchymal transition, and mitochondrial DNA genotoxicity [133]. Data from the literature suggest that some metals, such as Ni, Cd, Cr, and As detected in the e-cig mainstream, can play a key role in lung cancer etiology or contribute to its progression [53].

### 5.1. Nickel

Evidence of increased lung cancer risk in Ni high-exposure cohorts is unequivocal, with a 3–4-fold excess. Furthermore, a significant increase in nasal sinus cancer that is quite rare in the general population has been reported [134]. These data are of particular interest considering that some e-cig devices release a higher concentration of Ni compared to CCs [132]. Nickel ions are able to induce heterochromatinization by binding to DNA–histone complexes and initiating chromatin condensation. Nickel compounds can produce histone hyperphosphorylation (H3S10), hypermethylation (H3K4), and hyperubiquitination (H2A and H2B), inducing epigenetic effects that can act on gene expression. Some authors hypothesized that the Ni-condensing effect on chromatin could be followed by DNA methylation, leading to cell transformation, and tumor progression and oncogenesis, if these silenced chromatin regions contained tumor-suppressor- or senescence-related genes. Interestingly, nickel sulfide induces the hypermethylation of the p16 suppressor gene, leading to malignant histiocytomas in a mice model [134]. In addition, Ni may interfere with the microRNA network to degrade mRNA or block protein synthesis [134]. Since the dysregulation of DNA methylation and histone modifications are key epigenetic mechanisms implicated in tumor initiation, cancer progression, and metastasis, exposure to Ni, along with toxic aldehyde and PAHs recorded in the e-cig mainstream, represents a non-negligible risk factor for cancer incidence. Furthermore, recent data from in vitro studies show how Ni exposure suppresses the expressions of GST-M2 through epigenetic regulation. The GST-M2 enzyme is involved in cellular detoxification processes, and it is associated with cancer in multiple ways, including the inhibition of tumor metastasis and invasion in lung cancer. It also protects against DNA damage and exhibits high methylation levels in human lung tumor tissue [135].

### 5.2. Cadmium

It is known that smoking CCs and ENDSs are some of the most common sources of Cd exposure [136], data from clinical trials indicate that patients who live in highly polluted zones suffered from an increased prevalence of lung cancer, and the prognosis worsens in patients with higher urinary concentrations of Cd [137]. Cd increases oxidative stress which, in turn, contributes to lung inflammation with irritation of the respiratory tract and lung edema [138]. Interestingly, an animal model shows how Cd exposure induces genotoxicity via DNA strand breaks, mutations, chromosomal damage, impaired DNA repair, and cell transformation [139].

### 5.3. Chromium

Cr occurs mostly in tri (III)- and hexavalent (VI) states in the environment. Occupational exposure to Cr (VI) compounds is a well-documented cause of respiratory cancers, whereas Cr (III) was initially considered a relatively nontoxic agent [93]. However, recent evidence suggests that both tri- and hexavalent states can significantly induce genetic mutations in yeast and cause DNA damage [140], increasing mutational rates and causing DNA degradation. Cr (III) can bind DNA and irreversibly destroys the molecule, and it can interfere with the stacking mode of the base pairs. These results highlight the health relevance of the presence of Cr in e-cigs.

### 5.4. Arsenic

Another heavy metal detected in ENDSs is As, which is a highly toxic category I carcinogen metal, and the lung represents the main target organ [141]. As exposure can occur through ingestion and inhalation, respectively, in the form of soluble arsenite and particulate arsenic trioxide. Soluble arsenite has been proven to induce lung tumors [142], and various studies suggest that exposure to As during gestation and childhood could result in lung cancer in adulthood [143]. Tumor-suppressive miR-218-5p has a known role in regulating cancer proliferation, migration, and angiogenesis and recent evidence has shown that As exposure inhibits miR-218-5p. In addition, miR-218-5p specifically targets EGFR, regulating its expression, and the downregulation of this signaling pathway results in an overexpression of EGFR, a known lung cancer biomarker [144]. Chronic As exposure is also associated with an upregulation of programmed death-1 (PD-1)/programmed death-ligand 1 (PD-L1) (PD-1/PD-L1), increased regulatory T cells, and decreased CD8 cells as an anti-tumor immunosuppressor, along with higher levels of oxidative stress and DNA damage in lung epithelial cells via strand breaks [145,146].

## 6. Conclusions

Due to their recent release on the market, epidemiological data on the toxicological risk associated with the long-term use of e-cigarettes are not yet available. However, a growing number of preclinical evidence indicates that their use could represent a risk factor for several chronic pathologies, including cancer. The present article reviews the metal carcinogenesis risk associated with e-cig consumption. Both e-cig and HTP mainstreams present toxic metals at higher concentrations than those recorded in CC smoke. Cd, Ni, Pb, and Cr are the most common and they have been detected in different e-cig models/brands. Importantly, the exposure to these metals is associated with an increased cancer risk at different sites. The metals can, in fact, act as ROS generators by affecting the mitochondrial function and redox reactions in biological systems, which in turn lead oxidative DNA damage, and the deregulation of several cellular signaling pathways involved in cell cycle control and inflammation, which are common mechanisms of carcinogenesis.

Furthermore, by contrast to CCs, ENDSs, due to their many differences in terms of manufacturing, materials, and liquid composition, present significant differences in toxic compounds released, especially metals. Importantly, the customization of these devices by the consumers (battery output, temperature, and atomizer setting) generates different amounts of toxics and significantly affects the leaching of metals.

To date, we are far from definitively identifying the health impact of e-cigarettes and their putative role as smoking cessation support. As recently reiterated by the World Health Organization, both tobacco products and e-cigs pose risks to health, and we have no strong evidence supporting the harmless nature of electronic devices compared to tobacco smoke. More research should be conducted to clarify the content of these metals in the liquids of e-cigs and in the tobacco sticks of HTPs, as well as the mainstream they generate, in order to give a better understanding of potential metal exposure and subsequent implications.

## Figures and Tables

**Figure 1 ijms-25-02737-f001:**
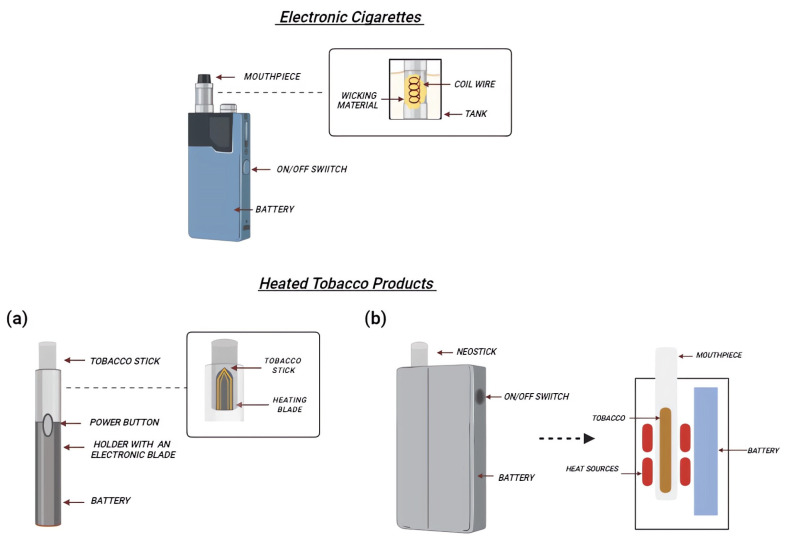
Schematic representation of e-cig and HTP components. E-cigs are characterized by a refillable tank (holding the e-liquid), an atomizer (activated by a button to aerosolize the e-liquid), and the battery. In HTPs, the disposable tobacco stick is heated thanks to an electronic blade, manually activated through a button (**a**). Alternatively, other devices use an induction heating chamber, where tobacco sticks are inserted into it (**b**).

**Figure 2 ijms-25-02737-f002:**
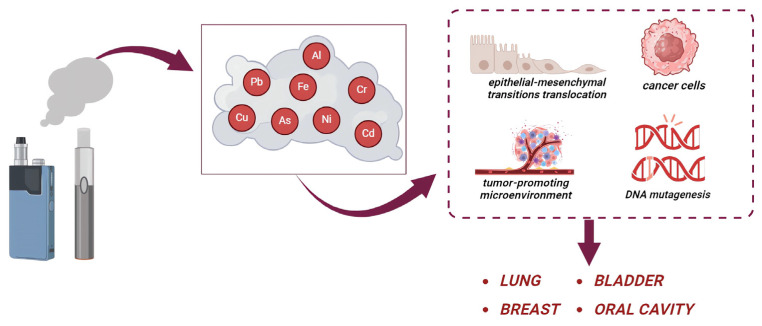
Effects of heavy metals released by e-cigarettes. The users are exposed to many heavy metals (such as As, Cd, Cr, Ni, and Pb). These metals participate in the development of cancer, by increasing the epithelial–mesenchymal transition translocation, tumor-promoting microenvironment, and DNA mutagenesis leading to lung, breast, bladder, and oral cancer.

**Figure 3 ijms-25-02737-f003:**
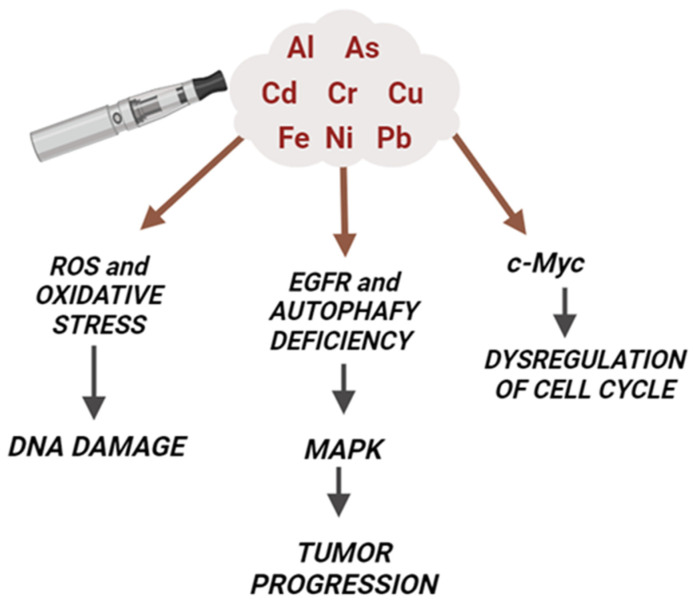
Molecular mechanisms of ENDS-induced cancer and the contribution of metals. The presence of many heavy metals (such as As, Cd, Cr, Ni, and Pb) in the ENDS mainstream contributes to the carcinogenic effects of these devices. In particular, metals are able to increase the production of ROS and oxidative stress, leading to DNA damage as well as activating cancer cell markers (like EGFR and MAP kinases) and oncogenes (c-Myc).

**Table 1 ijms-25-02737-t001:** Average content of heavy metals in tobacco, CC smoke, e-cig, and HTP mainstream.

	Metals Content in Tobacco (μg/g, Dry Weight)	Rate of Metals Transfer from Tobacco to CC Smoke (%)	Metals Content HTP Mainstream (μg/Stick)	Metals Content in E-Cig Mainstream (ng/m^3^)	Refs.
As	0.14	33–44%	<0.36	-	[51,52,54]
Cd	0.79	69–85%	<0.28	0.01	[51,52,54]
Cr	1.39	20%	<15.9	6.86	[51,52,54]
Ni	2.10	56.3 65.5%	-	0.30	[51,52,54]
Pb	0.55	54.3–67.1%	2.23	0.19	[51,52,54]

**Table 2 ijms-25-02737-t002:** Concentrations of VOCs and PAHs in the mainstream of the HTP IQOS and tobacco smoke. The data reported in the table compare the contents of the HTP IQOS mainstream with the contents of conventional cigarettes (Lucky Strike Blue Lights).

VOCs	HTPs (μg/Stick)	CCs (μg/Cigarette)	PAHs	HTPs (μg/Stick)	CCs (μg/Cigarettes)	Ref.
Acetaldehyde	133	610	Naphthalene	1.6	1105	[7]
Acetone	12.0	95.5	Acenaphthylene	1.9	235	[7]
Acroleine	0.9	1.1	Acenaphtene	145	49	[7]
Benzaldehyde	1.2	2.4	Fluorene	1.5	371	[7]
Crotonaldehyde	0.7	17.4	Anthracene	0.3	130	[7]
Formaldehyde	3.2	4.3	Phenanthrene	2.0	292	[7]
Isovaleraldehyde	3.5	8.5	Fluoranthene	7.3	123	[7]
Propionaldehyde	7.8	29.6	Pyrene	6.4	89	[7]
			Benz[a]anthracene	1.8	33	[7]
			Chrysene	1.5	48	[7]
			Benzo[b]fluoranthene	0.5	24	[7]
			Benzo[k]fluoranthene	0.4	4.3	[7]
			Benzo[a]pyrene	0.8	20	[7]
			Indeno[1,2,3-cd]pyrene	-	-	[7]
			Benzo[ghi]perylene	-	-	[7]
			Dibenzo[a,h]anthracene	-	-	[7]

## Data Availability

New data were not created.

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
