# Peer review of "Toxicological Aspects Associated with Consumption from Electronic Nicotine Delivery System (ENDS): Focus on Heavy Metals Exposure and Cancer Risk"

_ijms, 2024, doi:10.3390/ijms25052737_

Round 1

Reviewer 1 Report

Comments and Suggestions for Authors

Overall this is a nice snapshot of the current landscape of metals in heat-not-burn and electronic cigarettes. 

Throughout the paper, combustible cigarettes are referred to as many different things ("cigarette", "conventional cigarette", "traditional cigarette", "tobacco cigarette", etc). Authors should choose a single naming convention and use that as it is confusing in its current state.

A few minor comments:

Page 2, line 49 - Reference 6 does not discuss free radicals at all. Authors should consider reviewing and citing Bitzer et al - "Effects of Solvent and Temperature on Free Radical Formation in Electronic Cigarette Aerosols" to be more accurate when talking about the breakdown of e-liquid solvents.

Page 3, line 35 - "(PMID: 29530840)." should be a citation

Page 4, lines 158-159 - It would be beneficial to expand a bit upon the impact of flavorings here as these are the primary component of a user's choice of ecig/heat stick. Consider reviewing flavor works by Bitzer ZT et al and S Salam et al.

Page 4, lines 180-1801 - This is not entirely true as not all HTP products utilize blades. There are devices like the Glo which consist of a thin tobacco stick (neo stick) and a heating device similar to an oven. Devices like this should be mentioned.

Page 10, Section 5 - A more comprehensive discussion on the metal levels found in combustible cigarettes as compared to ENDS would greatly improve the paper.

Comments on the Quality of English Language

There are a large number of very long, awkwardly worded sentences. For example, in the abstract (lines 16-20) - This is very difficult to read/understand.

An extensive edit with a focus on reducing extremely long sentences like this would greatly improve the overall readability.

Author Response

REVIEWER 1

Overall this is a nice snapshot of the current landscape of metals in heat-not-burn and electronic cigarettes. 

Throughout the paper, combustible cigarettes are referred to as many different things ("cigarette", "conventional cigarette", "traditional cigarette", "tobacco cigarette", etc). Authors should choose a single naming convention and use that as it is confusing in its current state.

We thank the reviewer for highlighting this, we fully agree and so have edited the manuscript to use conventional cigarette (CC) throughout.

A few minor comments:

Page 2, line 49 - Reference 6 does not discuss free radicals at all. Authors should consider reviewing and citing Bitzer et al - "Effects of Solvent and Temperature on Free Radical Formation in Electronic Cigarette Aerosols" to be more accurate when talking about the breakdown of e-liquid solvents.

We agree with the Reviewer and have now added the correct reference to the manuscript.

Page 3, line 35 - "(PMID: 29530840)." should be a citation

We apologise for this oversight and the PMID has now been changed to a proper citation.

Page 4, lines 158-159 - It would be beneficial to expand a bit upon the impact of flavorings here as these are the primary component of a user's choice of ecig/heat stick. Consider reviewing flavor works by Bitzer ZT et al and S Salam et al.

We thank the reviewer for this suggestion. We have now expanded this section in the manuscript and included these two important publications.

Page 4, lines 180-1801 - This is not entirely true as not all HTP products utilize blades. There are devices like the Glo which consist of a thin tobacco stick (neo stick) and a heating device similar to an oven. Devices like this should be mentioned.

We agree with the Reviewer and we have now included this point and added a new reference.

Page 10, Section 5 - A more comprehensive discussion on the metal levels found in combustible cigarettes as compared to ENDS would greatly improve the paper.

We now included in the introduction of section 5 the discovery of the heavy metal concentrations reported in the literature for tobacco cigarettes, e-cigs and HTPs.

Comments on the Quality of English Language

There are a large number of very long, awkwardly worded sentences. For example, in the abstract (lines 16-20) - This is very difficult to read/understand.

An extensive edit with a focus on reducing extremely long sentences like this would greatly improve the overall readability.

We agree and Professor Fairclough, a native English speaker, has now properly reviewed the manuscript and corrected the English.

Reviewer 2 Report

Comments and Suggestions for Authors

The article is interesting, and the subject is actual in the context of the increasing popularity of ENDs. However, I would have liked to read a more extensive presentation of the potential carcinogenic effects of heavy metals in ENDS. Section 2 is too large in comparison to the other sections of the manuscript.

My main suggestion for the authors is to expand the sections about heavy metals in ENDs and their role in cancer initiation, promotion, and progression,  and provide more details. There are other references that the authors can consult and strengthen their debate. I will only give some examples below (but these titles are merely a suggestion).

Aalami AH, Hoseinzadeh M, Hosseini Manesh P, Jiryai Sharahi A, Kargar Aliabadi E. Carcinogenic effects of heavy metals by inducing dysregulation of microRNAs: A review. Mol Biol Rep. 2022 Dec;49(12):12227-12238. doi: 10.1007/s11033-022-07897-x. Epub 2022 Oct 21. PMID: 36269534.

Parida L, Patel TN. Systemic impact of heavy metals and their role in cancer development: a review. Environ Monit Assess. 2023 May 30;195(6):766. doi: 10.1007/s10661-023-11399-z. PMID: 37249740.

Zhao, L.; Islam, R.; Wang, Y.; Zhang, X.; Liu, L.-Z. Epigenetic Regulation in Chromium-, Nickel- and Cadmium-Induced Carcinogenesis. Cancers 202214, 5768. https://doi.org/10.3390/cancers14235768

Author Response

The article is interesting, and the subject is actual in the context of the increasing popularity of ENDs. However, I would have liked to read a more extensive presentation of the potential carcinogenic effects of heavy metals in ENDS. Section 2 is too large in comparison to the other sections of the manuscript.

My main suggestion for the authors is to expand the sections about heavy metals in ENDs and their role in cancer initiation, promotion, and progression,  and provide more details. There are other references that the authors can consult and strengthen their debate. I will only give some examples below (but these titles are merely a suggestion).

Aalami AH, Hoseinzadeh M, Hosseini Manesh P, Jiryai Sharahi A, Kargar Aliabadi E. Carcinogenic effects of heavy metals by inducing dysregulation of microRNAs: A review. Mol Biol Rep. 2022 Dec;49(12):12227-12238. doi: 10.1007/s11033-022-07897-x. Epub 2022 Oct 21. PMID: 36269534.

Parida L, Patel TN. Systemic impact of heavy metals and their role in cancer development: a review. Environ Monit Assess. 2023 May 30;195(6):766. doi: 10.1007/s10661-023-11399-z. PMID: 37249740.

Zhao, L.; Islam, R.; Wang, Y.; Zhang, X.; Liu, L.-Z. Epigenetic Regulation in Chromium-, Nickel- and Cadmium-Induced Carcinogenesis. Cancers 2022, 14, 5768. https://doi.org/10.3390/cancers14235768

We thank the Reviewer for their suggestions. Now we delved deeper into the metals carcinogenesis mechanisms discussing the results presented in the papers highlighted.

Reviewer 3 Report

Comments and Suggestions for Authors

I read with much excitement and eagerness the review of the literature drafted by Silvia Granata et-al. The authors present up to date literature of translational studies that give significance concerns for the potential of ENDs to lead to malignancy in humans, although as acknowledged by the authors the epidemiological literature quite does not support that association yet. They focus on heavy metals present in the aerosol produced by ENDs, which could be heavier than in conventional cigarettes.

The material is very relevant and important as ENDS gain popularity around the. The literature presented is excellent and my concerns are mostly related to the formatting of the presentation which can be significantly improved.

Major comments:

There are multiple grammatical errors and redundancy in the draft that makes it tedious to read at times. The work needs to be revised by a native English speaker as obvious grammatical mistakes are easily identified.

The authors are experts in the field and recently published an excellent review titled “On the toxicity of e-cigarettes consumption: Focus on pathological cellular mechanisms” (reference 5) which is reference in this draft at least 7 times, which could be perceived as “self-promoting” behavior in this manuscript. Instead, I would favor having the original research referenced.

The authors should consider the following format for the outline of their manuscript:

1.       Introduction, where the definition of ENDS is presented highlighting the wide diversity of products existing in the market making it challenging to conduct human research. Detailing the characteristics of e-cigs and HTPs, how they devices work and differences on product consumed (e-juice vs sticks) and aerosol produced by them. This could/should also be summarized in a table for reader convenience and simplicity. Here they can mention toxicants found in the aerosol of ENDs and their users (VOCs, PAH, etc) and then mentioned why they are focusing on heavy metals.

2.       The Device. 2.a. E-cigarettes detailing how heavy metals can be produced (by the coils, solder, etc) and 2.b. HTPs (and ideally showing how they differed from conventional cigarettes).

3.       The consumables. 3.a. The e-juice and 3.b. the tobacco sticks (and differences with conventional cigarettes).

4.       Characteristics of the aerosol produced by 4.a e-cg and 4.b HTPs. Here you could also mentioned other carcinogenic such as PAHs, VOC, etcs.

5.       Heavy metals from ENDS. Describing the most commonly found and the ones with the highest levels. 5.a Nickel, 5.b Cadmiun, 5.c Crhomiun and 4.d Arsenic

6.       The potential mechanistic pathways in which heavy metals from ENDS can lead to malignancy. 6.a ROS/oxidative stress, 6.b EGFR/MAPK and 6.c c-Myc

7.       Conclusion and research recommendations.

Minor issues:

1.       Multiple misspellings throughout including in the abstract (line 25 wit the letter A missing in Arsenic).

2.       Line 51 mentions PAHs produced by HTPs but it was not mentioned that e-cigs also produced PAHs

3.       Provide a reference for this statement starting in Line 64 “…1950s, tobacco industries made efforts to reduce the toxicants, especially overall tar and nicotine yields of cigarette, through the introduction of filters in order to decrease their negative effects.”

4.       Provide a reference for the following statement: “Tobacco industries thus decided to evolve and develop new devices to deliver nicotine without the hazard of traditional cigarette smoke. In this way, Electronic Nicotine Delivery Systems (ENDs) were designed as a useful tool to help people to replace CCs.” Please keep in mind that in the US manufacturers and marketers presented the argument that e-cigs was for pleasure and not to “help” people (SOTTERA V. FDA/SMOKING EVERYWHERE V. FDA (2009)).

5.       Line 101 to 105 is redundant as this has been already mentioned in the introduction.

6.       Figure 1 shows a vape-pen which is practically considered obsolete with most user using mods e-cigs. A transaction of the devices showing how the e-juice contacts the wick and the coil may be helpful. This could also be depicted for HTPs showing the stick in contact with he heating blade and where the subject’s mouth enters in contact with the HTP (for the uninformed reader).

7.       The formatting for reference in line 135 needs to be fixed.

8.       The description of animal studies showing how voltage and resistance can affect exposure to toxicants is confusing.

9.       Perhaps explaining the difference between aerosol, vapor and smoke should be presented, as many times the authors use the terms interchangeably.

10.   Line 241 missing () for Se.

11.   A reference should be presented at the end of this sentence- line 249-: “Another study, sponsored as well by manufacturing companies, was about the analysis of hazardous compounds in the HTP smoke, compared to the CCs one”. And the author’s name should be correctly spelled: Amorós-Pérez in the same line.

12.   Line 292 use the word “vape” which was not introduced before and could be confused with the device itself or with the verb (action) of using e-cigarettes.

13.   Line 370 shows abbreviations GSH/GSSG which were not defined.

14.   Grammar: “In addition, the exposure to Cr activates EGFR in the lung tissue of animals and, also, in lung tumor tissue from nonsmoking people exposed to Cr with squamous lung carcinoma” in line 419.

Comments on the Quality of English Language

Multiple misspellings and grammatical errors encountered throughout the manuscript. At times repetitive sentences with redundant information.

Author Response

read with much excitement and eagerness the review of the literature drafted by Silvia Granata et-al. The authors present up to date literature of translational studies that give significance concerns for the potential of ENDs to lead to malignancy in humans, although as acknowledged by the authors the epidemiological literature quite does not support that association yet. They focus on heavy metals present in the aerosol produced by ENDs, which could be heavier than in conventional cigarettes.

The material is very relevant and important as ENDS gain popularity around the. The literature presented is excellent and my concerns are mostly related to the formatting of the presentation which can be significantly improved.

Major comments:

There are multiple grammatical errors and redundancy in the draft that makes it tedious to read at times. The work needs to be revised by a native English speaker as obvious grammatical mistakes are easily identified.

We agree and Professor Fairclough, a native English speaker, has now properly reviewed the manuscript and corrected the English.

The authors are experts in the field and recently published an excellent review titled “On the toxicity of e-cigarettes consumption: Focus on pathological cellular mechanisms” (reference 5) which is reference in this draft at least 7 times, which could be perceived as “self-promoting” behavior in this manuscript. Instead, I would favor having the original research referenced.

We agree with the Reviewer. It was not our intention, but we can understand that it might be misunderstood.

The authors should consider the following format for the outline of their manuscript:

  1. Introduction, where the definition of ENDS is presented highlighting the wide diversity of products existing in the market making it challenging to conduct human research. Detailing the characteristics of e-cigs and HTPs, how they devices work and differences on product consumed (e-juice vs sticks) and aerosol produced by them. This could/should also be summarized in a table for reader convenience and simplicity. Here they can mention toxicants found in the aerosol of ENDs and their users (VOCs, PAH, etc) and then mentioned why they are focusing on heavy metals.
  2. The Device. 2.a. E-cigarettes detailing how heavy metals can be produced (by the coils, solder, etc) and 2.b. HTPs (and ideally showing how they differed from conventional cigarettes).
  3. The consumables. 3.a. The e-juice and 3.b. the tobacco sticks (and differences with conventional cigarettes).
  4. Characteristics of the aerosol produced by 4.a e-cg and 4.b HTPs. Here you could also mentioned other carcinogenic such as PAHs, VOC, etcs.
  5. Heavy metals from ENDS. Describing the most commonly found and the ones with the highest levels. 5.a Nickel, 5.b Cadmiun, 5.c Crhomiun and 4.d Arsenic
  6. The potential mechanistic pathways in which heavy metals from ENDS can lead to malignancy. 6.a ROS/oxidative stress, 6.b EGFR/MAPK and 6.c c-Myc
  7. Conclusion and research recommendations.

We tried to improve the manuscript as much as possible following the suggestions of all of the Reviewer. We have therefore tried to make changes that were in line with the requests of all three Reviewers. As such, we have included two extra tables to try to present the differences in terms of heavy metals and toxics associated with different devices and traditional cigarettes.

Minor issues:

  1. Multiple misspellings throughout including in the abstract (line 25 wit the letter A missing in Arsenic).

We thank the Reviewer. We edited the manuscript.

  1. Line 51 mentions PAHs produced by HTPs but it was not mentioned that e-cigs also produced PAHs

We thank the Reviewer now this point was included.

  1. Provide a reference for this statement starting in Line 64 “…1950s, tobacco industries made efforts to reduce the toxicants, especially overall tar and nicotine yields of cigarette, through the introduction of filters in order to decrease their negative effects.”

We thank the Reviewer. We edited the manuscript in accordance with the suggestion.

  1. Provide a reference for the following statement: “Tobacco industries thus decided to evolve and develop new devices to deliver nicotine without the hazard of traditional cigarette smoke. In this way, Electronic Nicotine Delivery Systems (ENDs) were designed as a useful tool to help people to replace CCs.” Please keep in mind that in the US manufacturers and marketers presented the argument that e-cigs was for pleasure and not to “help” people (SOTTERA V. FDA/SMOKING EVERYWHERE V. FDA (2009)).

We thank the Reviewer. We edited the manuscript in accordance with the suggestion.

  1. Line 101 to 105 is redundant as this has been already mentioned in the introduction.

We agree.

  1. Figure 1 shows a vape-pen which is practically considered obsolete with most user using mods e-cigs. A transaction of the devices showing how the e-juice contacts the wick and the coil may be helpful. This could also be depicted for HTPs showing the stick in contact with he heating blade and where the subject’s mouth enters in contact with the HTP (for the uninformed reader).

We now uploaded a new figure following the suggestions of the Reviewer.

  1. The formatting for reference in line 135 needs to be fixed.

We edited the sentence.

  1. The description of animal studies showing how voltage and resistance can affect exposure to toxicants is confusing.

We tried to make this point clearer.

  1. Perhaps explaining the difference between aerosol, vapor and smoke should be presented, as many times the authors use the terms interchangeably.

This is a key point. We think that best is to refer to the e-cigarette emission as a “mainstream” as many authors usually do. This is because the composition of the emission varies greatly from device to device, and for HnB cigarettes, even if the manufacturers talk about aerosol, actually, as shown by various works and also reported in the tables included in this manuscript, chemical elements such as PAHs are present, which are the chemical signature of an incomplete combustion and therefore it should be referred to it as smoke.

  1. Line 241 missing () for Se.

We thank the Reviewer for the report.

  1. A reference should be presented at the end of this sentence- line 249-: “Another study, sponsored as well by manufacturing companies, was about the analysis of hazardous compounds in the HTP smoke, compared to the CCs one”. And the author’s name should be correctly spelled: Amorós-Pérez in the same line.

We agree with the Reviewer and have jade this correction

  1. Line 292 use the word “vape” which was not introduced before and could be confused with the device itself or with the verb (action) of using e-cigarettes.

Thank you, we have edited this sentence accordingly.

  1. Line 370 shows abbreviations GSH/GSSG which were not defined.

We thank the Reviewer for highlighting this, it has now been defined in the text

  1. Grammar: “In addition, the exposure to Cr activates EGFR in the lung tissue of animals and, also, in lung tumor tissue from nonsmoking people exposed to Cr with squamous lung carcinoma” in line 419.

We thank the Reviewer for their extensive work on the manuscript. We edited the manuscript in line with the above suggestions and Professor Fairclough, a native English speaker, has now reviewed the manuscript.

Round 2

Reviewer 2 Report

Comments and Suggestions for Authors

The authors did a good job revising their manuscript. I have no other comments about the content of the paper. 

Author Response

We are pleased to learn that our improvements to the manuscript met the Reviewer's requirements and we thank her/him for her/his work on our paper.

Reviewer 3 Report

Comments and Suggestions for Authors

Thank you for giving me the opportunity to review the revise manuscript.

Major comments:

I believe that there have been significant improvements in the presentation of the content which is rich and informative, and the changes in the graphics and the inclusion of tables make the manuscript more relevant. Presenting objective data on the measurements of heavy metals in cigarette smoke, and the aerosol produced by E-cigs and heat not burn products really enrich the review. A whole new section on epigenetic changes and gene expression, as well as microRNAs have been introduced which provide cutting edge information to the reader expanding on potential oncogenic pathways associated with the heavy metals present in the emission of these novel tobacco products.

 However, I am disappointed that the authors have not paid enough attention to detail and have not communicated enough about the manuscript which could raise issues with authorship. Furthermore. multiple grammatical errors remain. The manuscript is many times repetitive (even word by word sentences are repeated ie: "The low voltage was perceived as safer; however, in vitro models have demonstrated that maintaining the same voltage" which appears twice). I would also suggest that the authors review the manuscript for word “again”, “said previously” and reconsider rewriting the concepts presented to make it more succinct.  As the authors revised the manuscript changes in the definition of aerosol, smoke and “mainstream” were introduced but improperly used throughout the manuscript. Some of the references may not be relevant and when used multiple times is repeating previously introduced concepts that do not need to be repeated.

Minor comments:

There are too many to mention. A few examples relevant to the “final version without track changes”:

Line 65 – Reference 12 is not relevant to the sentence presented. Same concept presented in line 93 with the use of the same reference.

Sentence wrongly worded: “Line 105 - The most appealing feature is the possibility to customize these devices with different PG/VG ratios, different nicotine percentages, different aromas (mint, vanilla menthol, etc.), and with the modulation of voltage and resistance, which determines the levels of toxicants in the generated vapour”. Additionally, the word aromas should be changed to flavors for consistency with other literature.

Line 114 call the aerosol produced by e-cigs “smoke”, with the use of multiple terms throughout the manuscript.

Line 128 – One could argue that most the rapid popularity of the new devices has depended in new users who have never been smoker, thus the quick raise of use among teenagers and young adults.

Line 149 – The concentration of nicotine in the e-juice in the use exceeds 90 mg/ml in some products.

Line 157- now call aerosol the aerosol produced by e-cigs which is inconsistent throughout the manuscript.

Line 163 states that uses can add flavors, which had already been mentioned.

Line 166 mentions the formation of ‘harmful species’…. What species??? Free radicals I am assuming.

Line 179 Sentence starts with “As said previously….” No need to say it again

Line 186 uses the word “until” it should probably read “up to” 350 degrees.

Line 202. Consider modifying Figure 1 to illustrate the blade in the tobacco stick or the coil around the heating element for GloTM

Line 291 The misspelling of the authors was not corrected. It should read Amoros-Perez (not Perey based on reference 60.

Comments on the Quality of English Language

It needs significant editing. Consider hiring a professional to review grammar and improve the delivery of your message.

Author Response

I believe that there have been significant improvements in the presentation of the content which is rich and informative, and the changes in the graphics and the inclusion of tables make the manuscript more relevant. Presenting objective data on the measurements of heavy metals in cigarette smoke, and the aerosol produced by E-cigs and heat not burn products really enrich the review. A whole new section on epigenetic changes and gene expression, as well as microRNAs have been introduced which provide cutting edge information to the reader expanding on potential oncogenic pathways associated with the heavy metals present in the emission of these novel tobacco products.

 However, I am disappointed that the authors have not paid enough attention to detail and have not communicated enough about the manuscript which could raise issues with authorship. Furthermore. multiple grammatical errors remain. The manuscript is many times repetitive (even word by word sentences are repeated ie: "The low voltage was perceived as safer; however, in vitro models have demonstrated that maintaining the same voltage" which appears twice). I would also suggest that the authors review the manuscript for word “again”, “said previously” and reconsider rewriting the concepts presented to make it more succinct.  As the authors revised the manuscript changes in the definition of aerosol, smoke and “mainstream” were introduced but improperly used throughout the manuscript. Some of the references may not be relevant and when used multiple times is repeating previously introduced concepts that do not need to be repeated.

We thank the Reviewer for their second review of our manuscript. We have now further streamlined the text to make it easier to read and we have improved some figures to follow the suggestions raised.

Minor comments:

There are too many to mention. A few examples relevant to the “final version without track changes”:

Line 65 – Reference 12 is not relevant to the sentence presented. Same concept presented in line 93 with the use of the same reference.

We agree with the Reviewer and so we have removed the repetition

Sentence wrongly worded: “Line 105 - The most appealing feature is the possibility to customize these devices with different PG/VG ratios, different nicotine percentages, different aromas (mint, vanilla menthol, etc.), and with the modulation of voltage and resistance, which determines the levels of toxicants in the generated vapour”. Additionally, the word aromas should be changed to flavors for consistency with other literature.

Thank you for highlighting this, we have now edited the sentence.

Line 114 call the aerosol produced by e-cigs “smoke”, with the use of multiple terms throughout the manuscript.

We have now edited the manuscript and used the term “mainstream” to refer to what is emitted by the devices.

Line 128 – One could argue that most the rapid popularity of the new devices has depended in new users who have never been smoker, thus the quick raise of use among teenagers and young adults.

We agree with the Reviewer that the new devices are widely spread also among never-smokers. However, now this sentence has been removed in order to make the manuscript more succinct.

Line 149 – The concentration of nicotine in the e-juice in the use exceeds 90 mg/ml in some products.

Thank you for highlighting this, we have now edited the sentence.

Line 157- now call aerosol the aerosol produced by e-cigs which is inconsistent throughout the manuscript.

As mentioned above, in the present version of the manuscript we use the term mainstream to indicate what the electronic devices emit.

Line 163 states that uses can add flavors, which had already been mentioned.

Now we removed the sentence in order to make it more succinct and avoiding repetitions

Line 166 mentions the formation of ‘harmful species’…. What species??? Free radicals I am assuming.

We agree with the Reviewer now we edited the sentence in accordance.

Line 179 Sentence starts with “As said previously….” No need to say it again

We agree with the Reviewer so this has been removed.

Line 186 uses the word “until” it should probably read “up to” 350 degrees.

We agree with the Reviewer. This part is no longer present, it was removed during the reviewing process

Line 202. Consider modifying Figure 1 to illustrate the blade in the tobacco stick or the coil around the heating element for GloTM

We agree and have now we improved the figure 1 as suggested

Line 291 The misspelling of the authors was not corrected. It should read Amoros-Perez (not Perey based on reference 60.

We thank the Reviewer for noticing this error, this has now been corrected.

Round 3

Reviewer 3 Report

Comments and Suggestions for Authors

The manuscript is significantly improved and very much informative and easy to read/comprehend.

Minor suggestions:

Consider removing word systmes in line 50: "The ENDs ENDS systems contain elements that include the same harmful constitu- 50 as ENDS includes it.